# Bibliometric Review: Classroom Management in ADHD—Is There a Communication Gap Concerning Knowledge Between the Scientific Fields Psychiatry/Psychology and Education?

**Martina Dort [1],\*, Anna Enrica Strelow [1],\*,†, Blandine French [2], Madeleine Groom [2], Marjolein Luman [3], Lisa B. Thorell [4], Guido Biele [5] and Hanna Christiansen [1]**

1. Clinical Child and Adolescent Psychology, Department of Psychology, University of Marburg, 35032 Marburg, Germany; hanna.christiansen@uni-marburg.de
2. Division of Psychiatry & Applied Psychology, Institute of Mental Health, University of Nottingham, Nottingham NG8 1BB, UK; Blandine.French@nottingham.ac.uk (B.F.); Maddie.Groom@nottingham.ac.uk (M.G.)
3. Clinical Neuropsychology, Department of Clinical, Developmental and Neuropsychology, Vrije Universiteit Amsterdam, 1081 BT Amsterdam, the Netherlands; m.luman@vu.nl
4. Department of Clinical Neuroscience, Karolinska Institutet, 171 77 Stockholm, Sweden; lisa.thorell@ki.se
5. Norwegian Institute of Public Health, Division of Mental Health, 0473 Oslo, Norway; Guido.Biele@fhi.no
\* Correspondence: martina.dort@uni-marburg.de (M.D.); anna.strelow@uni-marburg.de (A.E.S.)
† Shared first authorship.

**Abstract:** Many students with ADHD experience educational attainment difficulties. Nevertheless, evidence-based classroom management strategies (CMS) are seldom used. This science–practitioner gap might be due to a lack of shared knowledge between the scientific fields of psychology/psychiatry and education. This review uses science mapping to explore the basis of the current stock of knowledge in each of the two scientific fields, compares current approaches, and examines whether implementation methods and related barriers are investigated topics. We conducted a systematic search of the literature to identify articles on CMS in ADHD. We then conducted co-citation analyses and bibliographic coupling analysis. The former revealed six clusters of psychology/psychiatry and five clusters of education. Bibliographic coupling analysis resulted in eight clusters, with literature from both fields. The majority of the research is conducted in the field of psychology/psychiatry; teachers' perspectives are focused only in the field of education. The number of studies on implementation and potential barriers is small. There was thus relatively little communication between the sciences, but the scientific fields have seemed to converge recently. Connecting the scientific fields more and concentrating on implementation methods and barriers is strongly needed to close the science–practitioner gap.

**Keywords:** ADHD; classroom management strategies; bibliometric review; communication gap; psychology/psychiatry; education

## 1. Introduction

About 5–7% of all students all over the world suffer from attention deficit/hyperactivity disorder (ADHD), meaning that approximately one in 20 students shows symptoms of inattention, hyperactivity, and impulsivity [1–3]. Those three ADHD core symptoms are visible in students' problematic classroom behavior, e.g., difficulty focusing attention, distractibility, lack of impulse control, losing things, and forgetfulness [4]. This is connoted to a high variety of educational problems in school, like poor

performances in math, spelling, and reading, higher rates of repeating a school year, suspensions, and school exclusions [5,6]. About 50–80% of students with ADHD show at least one educational learning or achievement problem and those problems are seen as the mediator between symptoms of ADHD and increased risk of delinquency in the future [7]. Consequently, there is a need for treatment that reduces the symptoms of ADHD and decreases the risk of educational problems and concomitant difficulties.

Previous research indicates that the symptoms of ADHD are managed effectively with medication, (cognitive) behavioral therapy, and a combination of both [8,9]. Both treatments show significant effects on the symptoms of ADHD but often fail to improve academic achievement [9]. Effective classroom management strategies (CMS) supporting students with ADHD and offering teachers tools to decrease the impact of ADHD symptoms are therefore an important aspect of treatment [10].

Currently, teachers might use two kinds of behavioral strategies [11]: (1) antecedent-based strategies, such as modifying task-assignment or the definition of classroom rules; (2) consequent-based strategies, such as using quiet reprimands and token reinforcement [11,12]. The self-regulation or self-management approach is a special form of consequent-based CMS that uses contingency management to improve the student's ability to compare his/her own behavior to an external standard regarding this behavior. Thereby, the student learns to monitor and control their own behavior, for example by checking the task a second time [11,13].

A meta-analytic review shows that the most effective CMS are consequent-based strategies, such as using quiet reprimands and token reinforcement [11], or self-regulation strategies depending on the study designs that were used [12]. Those strategies show impressive effect sizes in decreasing off-task and disruptive behavior in children with symptoms of ADHD and further increasing the behavior of the classmates as well [12]. However, the strong association between effect- and sample size of the analyzed studies and the absence of control groups in most studies suggests that the generalizability of the large effect sizes is uncertain.

Despite the evidence supporting consequent-based strategies, teachers often use antecedent-based strategies [14]. Support for this assumption comes from studies where teachers reported that they primarily use corrective behavioral strategies and only seldom other kinds of behavioral or self-management strategies [14–16]. Overall, at least one of three students with ADHD does not receive any support in the classroom in the form of CMS, and this number is even higher for older students as well as for girls [17]. Whereas primary school teachers declare to apply more strategies when teaching children with ADHD, secondary school teachers do not [18], indicating that older students with ADHD do not receive sufficient support in the form of CMS. This suggests that the potential of CMS to enhance academic attainment in ADHD is not being met, partly due to teachers using strategies that are less effective for ADHD and a lack of application of these strategies to older students with ADHD. There is thus a science–practitioner gap entailing that evidenced CMS are not used best in daily school life and are often not used for older students with ADHD.

A possible reason for this science–practitioner gap might be a lack of communication between the scientific fields of psychology/psychiatry and education, which in return has led to a lack of information regarding effective CMS techniques to educational practitioners (e.g., teachers, school counsellors) about how to effectively deal with children with ADHD. Perhaps the two scientific fields focus on different aspects of handling and treating ADHD and fail to integrate their findings. This review thus aims to investigate the exchange of knowledge between the scientific fields of psychology/psychiatry and education. Furthermore, the aims of the current review are to quantitatively assess the basis of the current stock of knowledge in each of the two scientific fields and to compare these findings as well as the current approaches in those two scientific fields. Additionally, this review aims to find out if implementation methods and related barriers are investigated topics in either scientific field. This is important because even where there is good scientific and applied knowledge of CMS for ADHD, still there may be barriers to implementing these strategies; it is therefore useful to know whether this is recognized and explored in either scientific field [17].

To achieve these aims we conducted a bibliometric review using a specific approach, science mapping, which is defined as a quantitative method to statistically analyze patterns that emerge in the publication, citation and use of literature [19]. Science mapping is one of the main uses of bibliometric reviews [20]. It concentrates on the flow of information in science or in other words the structure of internal science communication. Hence, it allows us to synthesize previous research by classifying literature meta-data (e.g., authors, citations, documents, keywords) into different clusters, to visualize them with maps, and to identify links in the literature. When this meta-data is analyzed, a pattern of the most predominant literature in a field (the intellectual knowledge) can be revealed [20,21]. A bibliometric review is therefore better suited to the aims of the current research than other review methods such as systematic reviews or meta-analyses, as it handles a big set of data and provides a context of previous or following literature.

We conducted a bibliometric review to address the following hypotheses:

1. The scientific fields of psychology/psychiatry and education hold different intellectual knowledge regarding the usage of CMS for ADHD.
2. There is a communication gap between the scientific fields of psychology/psychiatry and education in the current literature concerning the exchange of knowledge.
3. Literature across both fields fails to focus on how to implement CMS in schools, how to support the teachers, or how to overcome possible barriers when implementing the CMS.

## 2. Materials and Methods

### 2.1. Literature Search

We used The Social Sciences Citation Index®(SSCI), offered by *Web of Science* for data collection as it is the most popular database in the analyzed scientific fields. It is interdisciplinary, and offers all the meta-data (title, author, abstract, key words, references, journal, year) of the literature that is relevant for a bibliometric analysis. As usual for a bibliometric review, our search was limited to one database [20]. The database SSCI was searched in November 2019 with the Boolean operators presented in Table 1. The search terms were based on the comprehensive meta-analysis by Gaastra, Groen, Tucha, & Tucha [12], which included general currently relevant terms in the context of ADHD research. Additionally, important key words in the literature about CMS for students with ADHD were supplemented.

The search resulted in $N = 422$ results in the scientific field of psychology/psychiatry and $N = 143$ results in the scientific field of education with an overlap of 40 articles (7%) in a time period from 1900 to 2019. We only included the following document types: articles, book chapters, early access articles, and reviews. Thus, twelve results (3%) in the scientific field of psychology/psychiatry and thirteen (9%) in the scientific field of education were excluded due to representing different document types (e.g. meeting abstracts or letters). For the time period of 2015 to 2019 we identified $N = 151$ articles in the scientific field of psychology/psychiatry and $N = 76$ articles in the scientific field of education with an overlap of eleven studies (5%). The non-overlapping results were reduced from $N = 216$ to $N = 202$ after applying inclusion criteria.

**Table 1.** Boolean operator for the literature search on CMS and ADHD.

| Terms for ADHD | | | Terms for CMS | | | | Terms for Scientific fields | | | Term for Time Span * |
|---|---|---|---|---|---|---|---|---|---|---|
| TS = ("ADHD") ("AD/HD") ("attention deficit") ("hyperactive" *) ("hyperkine" *) ("externali" *) | OR OR OR OR OR | AND | TS = ("antecedent-based") ("antecedent based") ("consequen *-based") ("consequen *-based") ("self-management") ("self-management") | OR OR OR OR OR OR | AND | | SU = (Psychology) (Psychiatry) respectively | OR | AND | PY = (2015–2019) |
| | | | ("school ("classroom ("education * ("academic * ("teacher * | + | Intervent *") Manage *") train*") strateg*") treat*") program") therapy") | OR OR OR OR OR OR | SU = (Education & Educational Research) | | | |

Note: The combination of the terms for CMS marked with a "+" were permuted. * The term for the time span was only used to filter the current literature for the bibliographic coupling analysis.

## 2.2. Bibliometric Methods

As we mentioned above, we used science mapping for this bibliometric review. More precisely, we used co-citation analysis to investigate the first and bibliographic coupling analysis to investigate the second and third hypothesis.

### 2.2.1. Co-citation Analysis

To assess the basis of the current stock of knowledge about CMS for pupils with ADHD in the scientific fields of psychology/psychiatry and education, document co-citation analysis was used. This method is based on the assumption that the more frequently two documents are cited together, the more equal their primary topic. Therefore, it analyzes how often documents appear together in reference lists [22]. In this way, this method is suitable to detect the most important literature ("the intellectual knowledge") of fields [23,24]. To figure out whether the scientific fields of psychology/psychiatry and education hold different stocks of knowledge, we conducted a separate co-citation analysis for each of the fields and compared the findings. The differentiation between those fields relies on WOS' classification.

As it takes some time until literature is cited a few times and this could lead to a bias towards older publications, this method does not necessarily reflect the currently most important articles in a field. Therefore, bibliometric coupling is a good supplement to this method [24].

### 2.2.2. Bibliographic Coupling

To investigate the exchange of knowledge of the current approaches in the two scientific fields of psychology/psychiatry and education, bibliographic coupling was used. In contrast to co-citation analysis, this method analyzes how often two documents cite the same reference and thus indicates how similar these documents are [20,25]. This method can only be applied to literature within a five-year time period, as, over time, the similarity of reference lists changes when newer studies are available that can only be included by newer papers [23]. We conducted a bibliographic coupling analysis with the literature of both fields combined to examine whether this literature forms rather topic- or scientific field-related clusters.

The findings of the bibliographic coupling analysis were also examined to determine whether implementation methods and related barriers are investigated topics in either field.

## 2.3. Data Cleaning and Analyzing

Data was first explored with the package *biblioshiny* for *RStudio Version 1.2.5019*. *Microsoft Excel 2016* was then used to identify different spellings of, e.g., author names. These alternate spellings were then used to create a thesaurus to merge duplicates for data cleaning. The data was then examined based on the described bibliometric methods. We applied network analysis to the data (for better traceability, see Figures 3 and 4). In this method, units of measurement (cited) documents are represented by network nodes, and the connections between them are represented by network ties [20]. For the co-citation analyses, network ties represent the frequency of co-occurrence of two documents in a reference list, and for bibliographic coupling analysis, network ties represent the number of equally cited references of two documents.

*VOSviewer 1.6.13* was used for the network analyses [26,27]. It executes mapping, which represents positioning of the analyzed documents according to the strength of their relationship, and clustering, which represents the subsumption of similar documents simultaneously, and combines the visualization of the results in one map. The VOS mapping technique is closely related to the multidimensional scaling, which reveals a map in a low-dimensional space by placing more similar nodes closer to each other [20,27]. Clustering is carried out according to a weighted and parameterized variant of the modularity function that again represents an algorithm introduced by Newman and Girvan (2004) [27].

The results produced by *VOSviewer* were imported into *Gephi 0.9.2* to calculate the eigenvalue centrality of the nodes. This measure calculates centrality according to the number of a node's connections and their connections [28]. It was used to identify the most relevant documents and, ultimately, the main topics for each cluster.

Finally, all results were imported into *Microsoft Excel 2016* and ranked by cluster and eigenvector centrality.

## 2.4. Interpretation of the Clusters

To interpret the clusters content wisely, every publication in every cluster was represented using the full list of authors, the publication year, and the belonging abstract. Afterwards, a group of ten raters that are unrelated to the project noted the content of the clusters with regards to the publications. With this information, two authors (MD and AES) defined the clusters' main topics individually and compared their results afterwards to choose the final inscription.

## 3. Results

### 3.1. Descriptive Results

As illustrated in Figure 1, the number of publications related to CMS and ADHD has increased substantially in the scientific field of psychology/psychiatry from the year 2003 and onwards and in the scientific field of education from the year 2014. The majority of research is done in the United States of America in both fields (Figure 2). As already mentioned, the overlap of literature of the two fields amounted to 7% for the time span from 1900–2019 and 5% for 2015–2019.

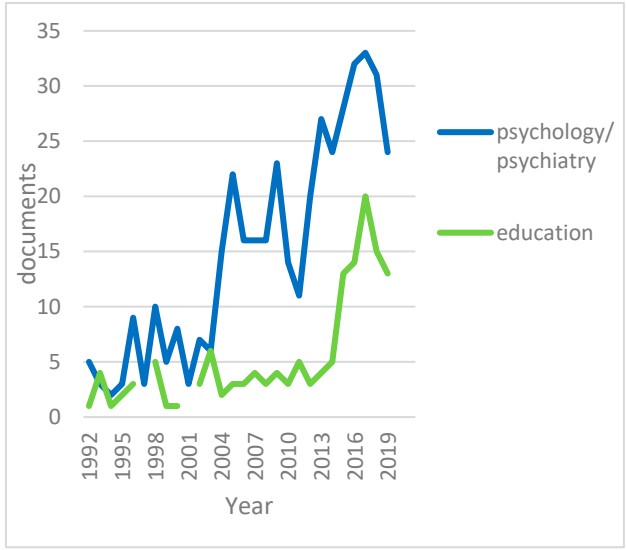

**Figure 1.** The number of published articles on CMS and ADHD in the scientific fields of psychology/psychiatry (*N* = 422) and education (*N* = 143) from 1992 to 2019. Breaks reflect no publications in these years. Note: three documents were published in 1977 and 1982, then there was a gap until 1992.

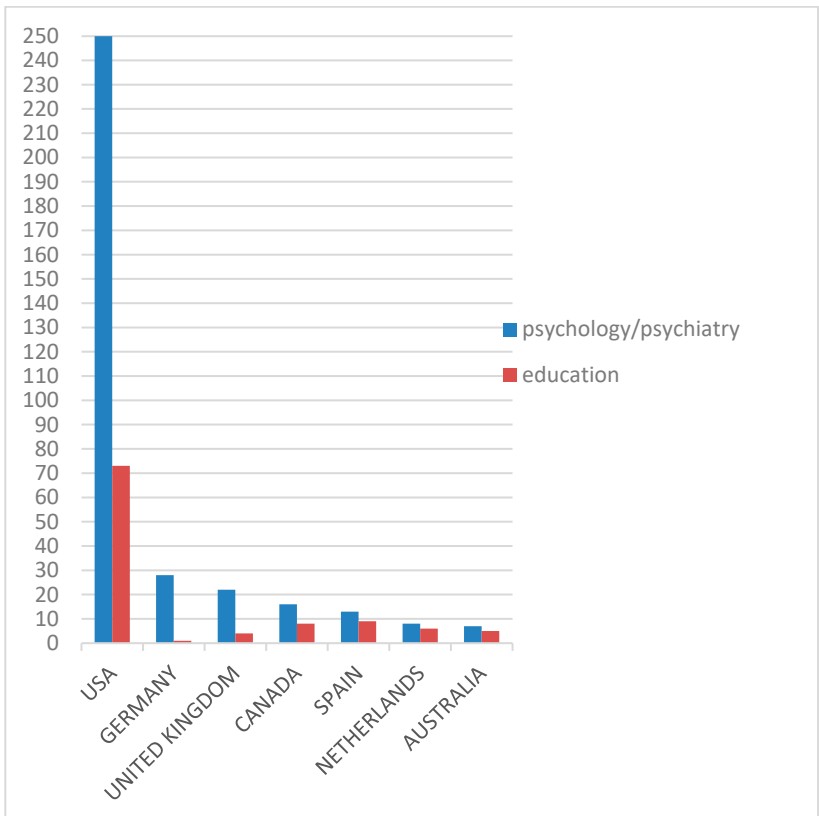

**Figure 2.** Countries of corresponding authors of the literature on CMS and ADHD in the scientific fields of psychology/psychiatry (*N* = 422) and education (*N* = 143). For each scientific field, the five most frequently occurring countries are presented.

*3.2. Co-citation Analyses*

Co-citation analysis revealed for the field of psychology/psychiatry 389 cited documents. Of these cited documents, 29% were assigned to a first cluster with the main topics *meaning of ADHD in the school* and *psychosocial treatment*; 25% to a second cluster including *medical, behavioral, and cognitive treatment* and *ADHD accompaniments*; 14% to a third cluster targeting *comparisons and combination of treatments* and *family-based factors of influence*; 14% to a fourth cluster with *academic functioning* and *research methods*; 12% to a fifth cluster with *academic performance* and *handling of disruptive behavior*; and 7% to a sixth cluster of *conception of ADHD over lifetime*. For the field of education, co-citation analysis revealed 166 cited documents, 36% of them in a first cluster with the main topics *teachers' perception and handling of challenging behavior*; 18 % in a second cluster with *evidence-based classroom interventions* and *single approaches*; 17% in a third cluster with *self-management and guidance for pupils with ADHD*; 15% in a fourth cluster with *teachers' knowledge and perception of ADHD* and *intervention effects*; and 13% in a fifth cluster with *diagnosis and treatment of ADHD according DSM IV and older*.

The related maps are illustrated in Figures 3 and 4, and detailed corresponding results are presented in Tables 2 and 3. The entire assignment of all documents to the clusters are available as *supplementary material—Bibliometric Review_Co-Citation_spreadsheet 1*.

Many clusters of both fields dealt with similar topics. Teacher-centered clusters only emerged in the field of education. Those clusters contained teachers' knowledge about ADHD as well as their current perception and practice of handling behavior problems in the classroom.

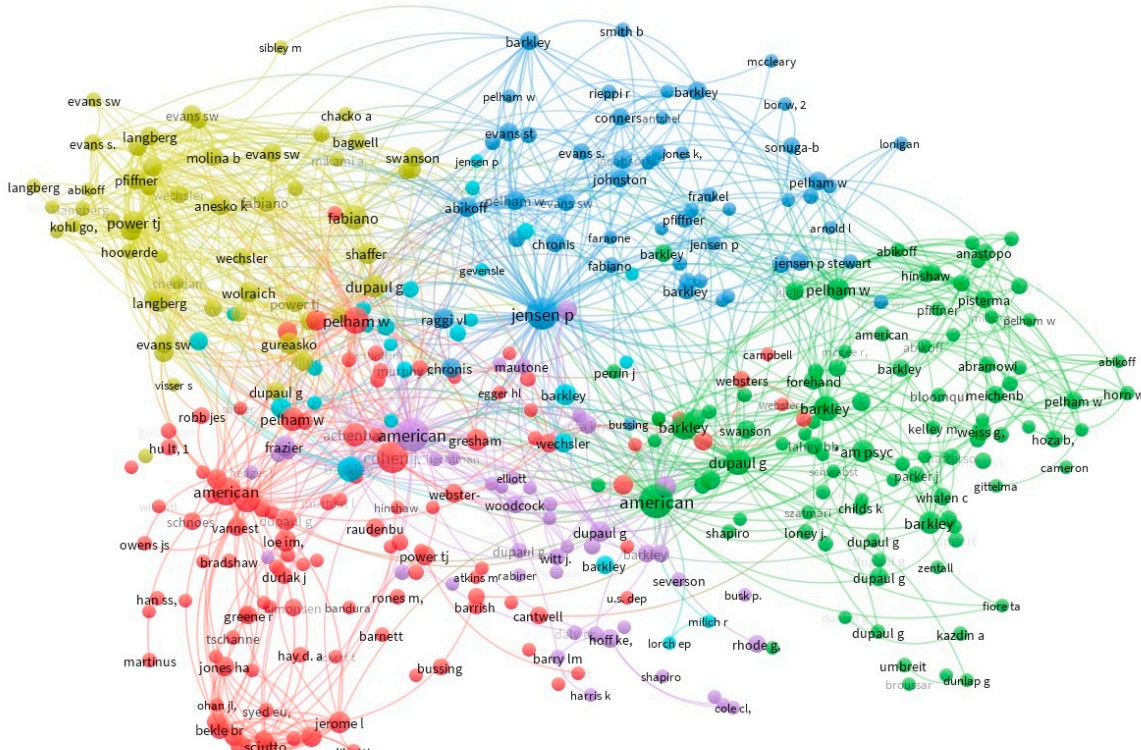

**Figure 3.** Co-citation analysis results represented as a combination of network maps and clustering for the scientific field of psychology/psychiatry. The analysis of *N* = 420 documents for the time period from 1900–2019 revealed 389 cited documents that subdivided into six clusters indicated by colors (see Table 2 for color codes and clusters' interpretation). Nodes represent cited documents and are labeled with the first author's name. Network ties represent the frequency of co-occurrence in a reference list with thicker lines reflecting higher co-occurrence. To reduce noise, lines were limited to 1000.

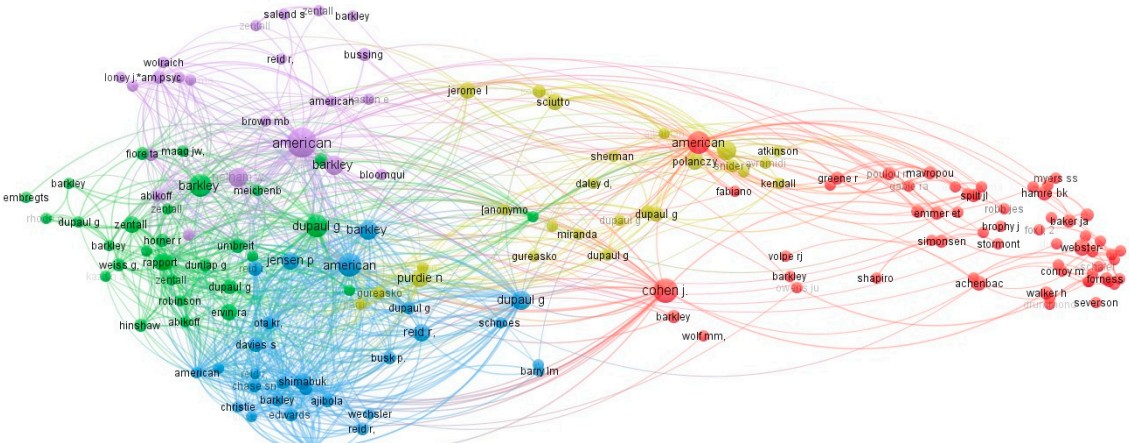

**Figure 4.** Co-citation analysis results represented as a combination of network maps and clustering for the scientific field of education. The analysis of *N* = 130 documents for the time period from 1900–2019 revealed 166 cited documents that subdivided into five clusters indicated by colors (see Table 3 for color codes and clusters' interpretation). Nodes represent cited documents and are labeled with the first author's name. Network ties represent the frequency of co-occurrence in a reference list with thicker lines reflecting higher co-occurrence. To reduce noise, lines were limited to 1000.

**Table 2.** Co-citation analysis results for the scientific field psychology/psychiatry. Analyzed was the co-citation of documents in the reference lists of *N* = 420 and documents in the time period from 1900–2019. Presented are the main topics of clusters. The color refers to the corresponding map in Figure 3.

| No. of Cited Documents | Main Topic(s) | Color |
| --- | --- | --- |
| 111 | meaning of ADHD in the school; psychosocial treatment | |
| 98 | medical, behavioral, and cognitive treatment; ADHD accompaniments | |
| 54 | comparison and combination of treatments; influential family-based factors | |
| 54 | academic functioning; research methods | |
| 46 | academic performance; handling of disruptive behavior | |
| 26 | conception of ADHD over lifetime | |

**Table 3.** Co-citation analysis results for the scientific field education. Analyzed was the co-citation of documents in the reference lists of *N* = 130 and documents in the time period from 1900–2019. Presented are the main topics of clusters. The color refers to the corresponding map in Figure 4.

| No. of Cited Documents | Main Topic(s) | Color |
| --- | --- | --- |
| 60 | teachers' perception and handling of challenging behavior | |
| 30 | evidence-based classroom interventions; single approaches | |
| 29 | self-management and guidance for pupils with ADHD | |
| 25 | teachers' knowledge and perception of ADHD; intervention effects | |
| 22 | diagnosis and treatment of ADHD according DSM IV and older | |

## 3.3. Bibliographic Coupling Analysis

The bibliographic coupling analysis resulted in *n* = 189 connected documents out of both fields and eight clusters, each of them with literature across both areas. The first cluster with the main topics *treatment effects and perception of ADHD* contained 22% of all connected documents; the second cluster *challenging behavioral problems* 18%; the third cluster *programs* and *importance of ADHD in schools, families, and society* 17%; the fourth cluster *self-efficacy* and *knowledge about ADHD* 16%; the fifth cluster *treatment effect studies* 13%; the sixth cluster *moderators of treatment effects* 8%; the seventh cluster *the role of parents/families* 5%; and the eighth cluster *implementation strategies* 2%.

A few general studies on implementation and potential barriers emerged across all clusters (e.g., [29–31]). The studies in the eighth cluster specifically concentrated on how to train practitioners. This cluster contained three studies, a very small number compared to the others. Detailed results are presented in Table 4 and the corresponding map in Figure 5. The entire assignment of all documents to the clusters are available as *supplementary material—Bibliometric Review Bibliographic coupling_spreadsheet 2*.

**Table 4.** Bibliographic coupling results for the scientific fields of psychology/psychiatry (psych.) and education for the time period from 2015–2019. Out of *N* = 202 analyzed documents, *n* = 189 were connected. Presented are the contents of the clusters. The color refers to the corresponding map in Figure 5.

| No. of Documents Total | No. of Documents Psych. | No. of Documents Education | Main Topic(s) | Color |
| --- | --- | --- | --- | --- |
| 41 | 23 | 18 | treatment effects and perception of ADHD | |
| 34 | 14 | 20 | challenging behavioral problems | |
| 32 | 24 | 8 | programs; importance of ADHD in schools, families, and society | |
| 30 | 23 | 7 | self-efficacy; knowledge about ADHD | |
| 24 | 19 | 5 | treatment effect studies | |
| 15 | 12 | 3 | moderators of treatment effects | |
| 10 | 10 | – | the role of parents/families | |
| 3 | 2 | 1 | implementation strategies | |

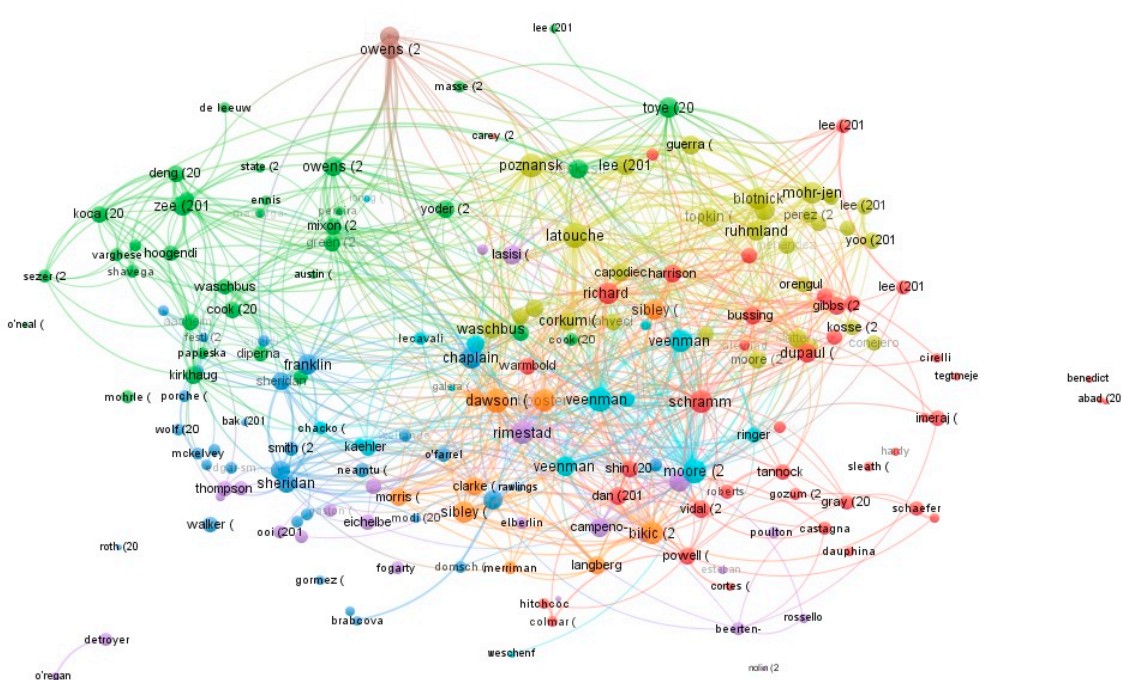

**Figure 5.** Bibliographic coupling results represented as a combination of network maps and clustering for the scientific fields of psychology/psychiatry and education. The analysis of *N* = 202 documents for the time period from 2015–2019 revealed *n* = 189 connected documents that subdivided into eight clusters indicated by colors (see Table 4 for color codes and clusters' interpretation). Nodes represent documents and are labeled with the first author's name. Network ties represent the number of references cited by both documents. To reduce noise, lines were limited to 800.

## 4. Discussion

### 4.1. Study Results

Previous research has shown that students with ADHD benefit from specific CMS that teachers can use to handle the symptoms of ADHD in the classroom. However, despite this, CMS are currently not used regularly in schools with the result that students often do not receive the support they could have [12,14–18]. We assumed that this science–practitioner gap exists because of a communication-related science–science gap between the scientific fields of psychology/psychiatry and education. Therefore, the initial aim of this review was to reveal and compare the structure of both scientific fields. Additionally, we assessed if the corresponding literature already discusses possible implementation barriers.

Our research shows a rising interest in the topic since 2003, which is presumably relatable to the *No child left behind Act* (NCLBA) that passed U.S. legislation in 2002 [32]. Not surprisingly, the primary research in this field is conducted in the U.S. A main requirement of the NCLBA was the implementation of standardized tests in schools in order to regularly assess the learning of each individual student. However, the NCLBA included only few suggestions for teachers as well as little requirements for a better classroom environment that would need educational research [32]. Therefore, it is not surprising that the majority of research in this area is done in the field of psychology/psychiatry, as this field typically deals with standardized assessments. In the scientific field of education, a significant increase

in publication can only be seen since 2014. This might be a rather long-term result of the NCLBA as this act did not only require a learning assessment for children with ADHD but also forced teachers to handle ADHD-related behavioral problems in the classroom, which might have resulted in increasing needs. Due to these new circumstances, the need for support for teachers might have grown, and with it research in the field of education.

### 4.1.1. Co-citation Analyses: Basis of Current Stock of Knowledge of both Fields

The first hypothesis states that the scientific field of psychology/psychiatry holds a different knowledge regarding the usage of CMS than the scientific field of education. The results found support for this hypothesis, in that there is only an overlap of 7% (40 publications in total) between the two fields from the year 1900 until 2019. From 2015 to 2019, the overlap is even less (5%, 11 publications in total). Co-citation analyses revealed six clusters in the scientific field of psychology/psychiatry and five in the scientific field of education, and those deal mainly with comparable topics. For instance, the topic of handling disruptive/challenging behavior (psychology/psychiatry = cluster 5, education = cluster 1) as well as the meaning of ADHD in school (psychology/psychiatry = cluster 1) or teachers' perception of it (education = cluster 4) occur in both fields. Furthermore, both fields examine the treatment of ADHD (psychology/psychiatry = cluster 1–3, education = cluster 2–4). The main difference in the intellectual knowledge is the focus on teachers' perspective and handling of ADHD-related behavioral problems in the educational scientific field, whereas the scientific field of psychology/psychiatry seems to have a multifaceted view on CMS and ADHD with, e.g., studies comparing different treatments. Furthermore, ADHD over the lifetime is not notably represented in the scientific field of education. In summary, the intellectual knowledge in the two fields does not merge and does not use a shared body of literature, though similar topics are handled. This should be considered to be a serious limitation and one of the most important ways of increasing our knowledge for a certain topic is that the new studies that we design build on the knowledge obtained in previous studies with the aim to reduce the daily practice-gap between mental health and education.

The results of the present review suggest that the strategies used by teachers and how teachers and psychologists/psychiatrists can work together effectively are neglected topics in the field of psychology/psychiatry. A similar picture appears in a previous review that aimed to investigate the effectiveness of psychoeducation for teachers and parents. This review only found one study that included teachers who received psychoeducation by another health professional [33]. Of course, CMS for students with ADHD is a topic that has to be investigated by the scientific field of education. At least since the release of the German S3-guideline for the treatment of ADHD in 2017, in which it is unambiguously claimed that psychosocial treatments are necessary in the treatment of ADHD, this paradigm should have changed [34]. Even the U.S. National Institute of Mental Health (NIMH) claims on its public webpage that CMS need to be part of treating ADHD [35]. However, these are new guidelines, and a limitation of co-citation analyses is that it takes some time before publications are cited. This is the reason why we choose to complement this data analysis with biographic coupling as discussed below; thus, this method is not suitable to detect recent changes in the literature, which leads us directly to the second hypothesis.

### 4.1.2. Current Research Approaches

Our second hypothesis assumes a communication gap between the scientific fields of psychology/ psychiatry and education in the recent literature. This hypothesis does not hold true according to results of bibliographic coupling. In all eight clusters, psychological and educational literature is represented, except for the seventh cluster (*the role of parents/families*), which is only present in the psychology/psychiatry field. Additionally, the clusters are close to each other, indicating a high correspondence. There are two aspects that should be highlighted: The second cluster (*challenging behavioral problems*) includes more educational literature and is distant from the other clusters. Additionally, as mentioned previously, in the seventh cluster (*the role of parents/families*) no

educational literature is included. Interaction with families about CMS is difficult [18], and poor parent–teacher communication is common for children with ADHD [17]. We see that the fields seem to converge recently, but there are still topics that need more attention, such as implementation strategies that will be discussed in the next paragraph. In conclusion, psychologists/psychiatrists should focus more on the handling of challenging behavioral problems, as does the scientific field of education, so that the practitioners in this field know how to collaborate with teachers. Hopefully, a stronger focus on how to deal with challenging behaviors in the classroom within the field of psychology/psychiatry will result in stronger cooperation with teachers.

### 4.1.3. Investigation of Implementation Strategies and Barriers

The third hypothesis states that scientific literature fails to focus on how to implement CMS in schools, how to support the teachers, and how to overcome possible barriers when implementing the CMS [17]. The results of the present study find support for this hypothesis, as these topics are rarely handled in either psychology/psychiatry or education. This might be one factor contributing to the observed science–practitioner gap. There is research about ADHD, as well as about CMS, and how efficacious they are in handling the symptoms of ADHD, but very few research papers on how to implement such strategies in the classroom and the potential barriers.

### *4.2. Implications*

With this bibliometric review, we are able to set a reference frame and to reveal blind spots of (current) research literature. The interest in the topic of CMS in conjunction with ADHD is rising; however, most research is done in the U.S. More research in different countries is needed, as school systems differ. Though the scientific fields of psychology/psychiatry and education have different knowledge bases that affect the relevant literature, we can see that the handled topics are comparable. The communication between the scientific fields should be improved, otherwise the potential to help students with ADHD is undermined. Scientific activity could be seen as a metaphor for what can happen between psychotherapists/psychiatrists and teachers in real life—both can do good work, but do not exchange key learnings or major issues, so that their potential to support an impaired child is limited.

Even if according to the results of the bibliographic coupling analysis both fields seem to have convergences, in psychology/psychiatry, a greater focus should be on teachers and their role in the treatment of ADHD. Health professionals should research how to advise and interact with teachers, so that an all-encompassing treatment can be developed. Furthermore, in the field of education, more interest in ADHD over the lifetime would be helpful, as ADHD is a disorder that often affects people throughout life and is therefore as relevant for secondary school students as for primary school students. In order for interventions to have a sustainable effect, it should be considered important for future research to take research findings from psychology, psychiatry, as well as education into both scientific fields' consideration.

Specifically, research should focus on how to implement CMS in schools. As we know from previous research, pre- and in-service teachers are lacking knowledge about ADHD, and such a lack proved to be the most important variable influencing teachers' attitudes towards effective and ineffective CMS in handling problematic behavior due to ADHD as well as their intention to use them [15,16]. Not only researchers, but also every practitioner of both fields should have that in mind and should find ways to promote a better exchange and to impart knowledge to one another. For example, it would be helpful for students with ADHD if more consequent-based instead of antecedent-based and corrective behavioral strategies would be used by teachers [14–16]. For instance, to put self-regulation strategies for students with ADHD into practice, a regular exchange between teachers and psychotherapists would be very helpful, as this allows an exchange of ideas about the individual student's needs, resources, and problems [12].

Literature about how to implement CMS and overcome barriers is rare. CMS need more research attention as teachers might know about CMS, but they do not seem to receive support for implementing such strategies and might be unclear how to adjust implementation strategies to optimize CMS.

*4.3. Limitations*

With this review, we demonstrate a significant gap between the scientific fields of psychology/ psychiatry and education regarding their shared knowledge. We assume that this science–science gap is part of the problem of the science–practitioner gap, i.e., effective classroom interventions not being implemented in schools. Of course, this lack in previous science communication is not the exclusive explanation for the observed science–practitioner gap, but it might be a first step to overcome it if more research is focused on both perspectives, as this might increase the overlap in knowledge between fields.

Bibliometric reviews have some strengths compared to other kinds of reviews, such as handling a big set of data and providing a context of literature, but also face some limitations: First, some literature is cited very often, because it is relevant for a specific method, but often this literature has nothing to do with the searched topic (e.g., Cohen [36] is often cited with reference to effect sizes, though this citation has nothing to do with the researched topic). Secondly, the meta-data we use might be biased, as it is gathered by individuals. To limit this, we double reviewed it. Third, it is not possible to filter noise produced by literature that does not reflect the researched topic but complies with the literature search criteria. Thus, we were not able to include the search term "ADD" (attention deficit disorder) in addition to ADHD in our research, because it would be identical to the word "add" and therefore would have produced too much noise.

**5. Conclusions**

The similarity of both scientific fields seems to have increased over the last years, but there are still aspects that need to be ameliorated so that the science–practitioner gap might get closed. First, more research conducted outside the U.S. is needed, because not all findings are transferable from the U.S. to other countries, due to differences in the school systems. Additionally, the scientific field of education needs to pay more attention to ADHD over the lifetime so that not only younger but also older students receive the support they need. The scientific field of psychology/psychiatry should focus more on the cooperation with teachers when reporting about the treatment of ADHD. Moreover, both fields should concentrate on how to implement CMS in schools, how to train teachers, and how to handle potential barriers.

**Supplementary Materials:** The following are available online at http://www.mdpi.com/2071-1050/12/17/6826/s1: Bibliometric Review_Co-Citation_spreadsheet 1; Bibliometric Review_Bibliographic Coupling_spreadsheet 2.

**Author Contributions:** Conceptualization: M.D., A.E.S., and H.C.; methodology: M.D., A.E.S., and H.C.; validation: H.C.; formal analysis: M.D. and A.E.S.; investigation: M.D. and A.E.S.; resources: H.C.; data curation: M.D., A.E.S., and H.C.; writing—original draft preparation: M.D. and A.E.S.; writing—review & editing: H.C., B.F., M.G., M.L., L.B.T., and G.B.; visualization: M.D.; supervision: H.C.; project administration: H.C., funding acquisition: H.C. All authors have read and agreed to the published version of the manuscript.

**Funding:** This research was part of the project "ADHD in the classroom" that is part of the RTG 2271 and funded by the German Research Foundation (DFG) project number 290878970-GRK 2271, project 1.

**Conflicts of Interest:** The authors declare no conflict of interest.

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
