# Peer review of "Bibliometric Review: Classroom Management in ADHD—Is There a Communication Gap Concerning Knowledge Between the Scientific Fields Psychiatry/Psychology and Education?"

_sustainability, doi:10.3390/su12176826_

Round 1
Reviewer 1 Report
Thank you for the possibility to review this manuscript. The authors examine the timely and important question regarding the implementation of extensive research findings in the ADHD field into practice. The manuscript is appropriately segmented and comprehensive. My only suggestion is to elaborate more on how the education field could use the research findings in psychiatry/psychology (Implications section).
Other minor: citation, line 168
Author Response
Thank you for the auxiliary and friendly comments. We added more recommendations in the implication section (lines 377 – 391). Furthermore, we changed the citation in line 168 (now 169) so that it is in accordance with the journal’s guidelines.

Reviewer 2 Report
Dear Authors,
Thank you for taking up a very important topic of Classroom Management in ADHD by analyzing and comparing the perspectives of two research fields and using the bibliometric review method.
The aim of the study has been fully achieved, the research methods have been selected and used correctly, and the results and conclusions have been clearly presented.
I have no critical comments about the text. You can only specify in the title that it concerns the fields of psychology/psychiatry and education, e.g.: „Bibliometric Review: Classroom Management in ADHD – Is There a Communication Gap Concerning Knowledge Between Psychology/Psychiatry and Education?
It is only a proposal for consideration, and I leave the final decision to the authors.
It was a pleasure to review such as interesting paper. I am pleased to recommend the text for publication.
Regards,
The reviewer.
Author Response
Thank you for this friendly and helpful feedback. We changed the title to ‘Bibliometric Review: Classroom Management in ADHD – Is There a Communication Gap Concerning Knowledge Between The Scientific Fields Psychiatry/Psychology and Education?’ due to this comment.
